# Impact of Postoperative Changes in Brain Anatomy on Target Volume Delineation for High-Grade Glioma

**DOI:** 10.3390/cancers15102840

**Published:** 2023-05-19

**Authors:** Cas Stefaan Dejonckheere, Anja Thelen, Birgit Simon, Susanne Greschus, Mümtaz Ali Köksal, Leonard Christopher Schmeel, Timo Wilhelm-Buchstab, Christina Leitzen

**Affiliations:** 1Department of Radiation Oncology, University Hospital Bonn, 53127 Bonn, Germany; 2Faculty of Medicine, University Bonn, 53127 Bonn, Germany; 3Department of Radiology, University Hospital Bonn, 53127 Bonn, Germany; 4Department of Radiology, Waldkrankenhaus, 53177 Bonn, Germany; 5Radiation Oncology Bonn-Rhein-Sieg, 53115 Bonn, Germany

**Keywords:** brain tumour, high-grade glioma, glioblastoma, radiation therapy, target volume delineation, radiation treatment planning, magnetic resonance imaging

## Abstract

**Simple Summary:**

Malignant brain tumours have a poor prognosis and routinely require brain surgery followed by radiation treatment of the extended tumour cavity. We compared magnetic resonance images (MRIs) of 28 patients at two points in time: immediately after surgery and close before the start of radiation. Even though both MRIs were roughly only 3 weeks apart, we noted substantial differences in the position and size of the tumour cavity, surrounding affected brain tissue, the position of the midline, and bleeding in the surgical area. The brain anatomy, as seen in the MRI, guides the planning of radiation. Older images thus do not reflect the actual anatomy anymore, which might lead to insufficient treatment of the tumour site or increased side effects due to irradiation of healthy tissues. Our data support the use of a second MRI for radiation treatment planning as close to its start as possible.

**Abstract:**

High-grade glioma has a poor prognosis, and radiation therapy plays a crucial role in its management. Every step of treatment planning should thus be optimised to maximise survival chances and minimise radiation-induced toxicity. Here, we compare structures needed for target volume delineation between an immediate postoperative magnetic resonance imaging (MRI) and a radiation treatment planning MRI to establish the need for the latter. Twenty-eight patients were included, with a median interval between MRIs (range) of 19.5 (8–50) days. There was a mean change in resection cavity position (range) of 3.04 ± 3.90 (0–22.1) mm, with greater positional changes in skull-distant (>25 mm) resection cavity borders when compared to skull-near (≤25 mm) counterparts (*p* < 0.001). The mean differences in resection cavity and surrounding oedema and FLAIR hyperintensity volumes were −32.0 ± 29.6% and −38.0 ± 25.0%, respectively, whereas the mean difference in midline shift (range) was −2.64 ± 2.73 (0–11) mm. These data indicate marked short-term volumetric changes and support the role of an MRI to aid in target volume delineation as close to radiation treatment start as possible. Planning adapted to the actual anatomy at the time of radiation limits the risk of geographic miss and might thus improve outcomes in patients undergoing adjuvant radiation for high-grade glioma.

## 1. Introduction

Glioblastoma (GBM) remains the most common malignant primary brain tumour in adults, with a global incidence of <10 per 100,000 people [1]. Standard of care includes maximal safe surgical resection followed by adjuvant radiotherapy plus concomitant and adjuvant temozolomide, which doubles survival chances [2,3]. Despite aggressive multimodality treatment, its prognosis remains dismal, with a median overall survival of around 15 months [2,4,5]. Continuous research efforts and innovative treatment options have brought only little improvement [3,4].

Nowadays, intensity-modulated radiation therapy (IMRT) is the gold standard, as it yields better target volume coverage while sparing important organs at risk (OARs) [6]. This potentially reduces treatment-related side effects and improves neurocognitive outcomes. To accurately delineate the target volume as well as the OARs, postoperative high-resolution magnetic resonance imaging (MRI) is used, as this allows for an accurate assessment of the cranial soft tissues. MRI has thus become the standard imaging modality for radiation treatment planning [7]. In these images, the gross tumour volume (GTV) is defined by the resection cavity and any T1-weighted contrast-enhancing residual tumour, if present. The clinical target volume (CTV) is generated by adding a 1−2 cm margin to the GTV while respecting the natural neuroanatomical borders (e.g., skull bone, falx cerebri, tentorium cerebelli, and ventricles) and includes any T2-visible peritumoral oedema, to account for microscopic tumour infiltration [7,8,9]: GBM is thought to be a systemic brain disease upon diagnosis, and autopsy studies have revealed subclinical low-grade tumour cell infiltration in oedema surrounding the resection cavity [10,11]. Despite these extended safety margins, local recurrences are common: more than two-thirds of GBM resurge within 2 cm of the tumour bed (in-field), making local tumour control essential [12]. Distant recurrences are less common, and extracranial metastases are rare [13].

Following surgery, a first MRI is usually performed within 24–48 h [7]. This serves several purposes: assessing the extent of resection (i.e., a prognostic factor), identifying any postoperative complications (e.g., ischaemia, bleeding, contusion, infection), and as a baseline for monitoring and detecting disease progression. Adjuvant radiotherapy is usually initiated within 3–5 weeks following surgery to allow for proper wound healing [7,14]. For radiation treatment planning, a repeat MRI is recommended to allow for an accurate and representative target volume and OAR delineation. There is, however, no consensus on when this imaging should be performed (usually no older than 1–2 weeks before radiation treatment start), and sometimes it is omitted altogether, e.g., due to limited availability or time or added costs. In these cases, the immediate postoperative MRI is used to guide contouring, even though short delays in radiation treatment initiation do not negatively impact survival [14].

The aim of this study is to compare the immediate postoperative MRI with the radiation treatment planning MRI and to determine the impact of postoperative changes in brain anatomy on target volume delineation for high-grade glioma.

## 2. Materials and Methods

### 2.1. Patient Selection

In this retrospective analysis, adults who previously received adjuvant radiation treatment for a malignant primary brain tumour at our university cancer centre between 2016–2017 were assessed for eligibility. Patients with unifocal disease, neuropathological confirmation of high-grade glioma, and an immediate postoperative brain MRI (<48 h following surgery; MRI 1) as well as a second MRI for radiation treatment planning (<10 days preceding the first fraction; MRI 2) were included. Patients with a history of prior cranial surgery or irradiation were excluded. This study was approved by the Institutional Review Board of the University Hospital Bonn (057/22).

### 2.2. MRI Protocol

All MRIs were performed on the same 3 Tesla devices (Ingenia 3.0T, Philips, Amsterdam, The Netherlands). An established “radiation treatment planning protocol” was used, which includes a T1-weighted contrast-enhanced (Gadolinium) sequence and a T2-weighted FLAIR (fluid-attenuated inversion recovery) sequence, both with a slice thickness of 1 mm.

### 2.3. Contouring

In order to delineate the volumes of interest, the Longitudinal Brain Imaging (LoBI) and Smart Region of Interest (ROI) tool of IntelliSpace Portal 11 (Philips, Amsterdam, The Netherlands) were used. Contouring was performed on both MRIs (1 and 2) by two senior radiologists. The resection cavity and any T1-weighted contrast-enhancing residual tumour (if present), as well as the surrounding oedema and FLAIR hyperintensities, were delineated separately, guided by the neuroradiological report. An example can be found in Figure 1.

The following data were extracted for both MRIs separately: location of the resection cavity, distance between the resection cavity border and skull bone (dorsal, ventral, lateral, medial, caudal, cranial), volume of the resection cavity, surrounding oedema, and FLAIR hyperintensities, presence and extent of midline shift, and presence and extent of subdural haematoma in the craniotomy region. Furthermore, patient and tumour characteristics, as well as the interval between MRI 1 and 2, were obtained.

### 2.4. Statistical Analysis

Mean, median, standard deviation (SD), and range were calculated for all applicable data. Data distribution was checked for normality with the Shapiro–Wilk test. For the comparison of independent continuous variables, the Mann–Whitney U test or Kruskal–Wallis test was used, depending on the number of samples. The statistical significance level was defined as *p* < 0.05, using R version 4.1.2 (*R* Foundation for Statistical Computing, Vienna, Austria) to perform the analyses.

### 2.5. Literature Search

A comprehensive search of international literature in the MEDLINE database was performed to identify similar cohorts, using PubMed as a primary search engine. Studies matching the search string *high-grade glioma AND radiation treatment volume AND magnetic resonance imaging* were screened for inclusion based on title and abstract. Additional studies were identified by cross-searching the already included articles’ references.

## 3. Results

### 3.1. Patient Characteristics

In total, 28 patients with high-grade glioma were eligible and included in the analysis. The median age (range) was 61 (26–78) years. GBM was the most common neuropathological diagnosis (93%), and two patients (7%) had oligodendroglioma. The tumour location was the frontal lobe in 46%, followed by parietal and temporal (25% each), and occipital lobes (4%). The median time (range) between MRI 1 and 2 was 19.5 (8–50) days. General patient characteristics are summarised in Table 1. A detailed overview including the MRI data is provided in Appendix A.

### 3.2. Changes in Resection Cavity Position

Overall, the mean absolute difference in the position of the resection cavity relative to the skull bone between MRI 1 and 2 was 3.04 ± 3.90 mm. Resection cavity borders were then divided into “skull-near” and “skull-distant”, depending on whether the closest distance to the skull bone was ≤ or >25 mm, respectively. For skull-near resection cavity borders, the mean absolute difference was 0.50 ± 1.40 mm, whereas, for skull-distant resection cavity borders, this was 4.61 ± 4.12 mm. This difference was statistically significant (*p* < 0.001), even after accounting for the interval between MRI 1 and 2. Data on resection cavity position are summarised in Table 2. Figure 2 shows a scatter plot of the individual relative positional changes between MRI 1 and 2.

The tumour location (frontal, parietal, temporal, occipital) did not influence the difference in position of the resection cavity between MRI 1 and 2, neither in skull-near (*p* = 0.069) nor in skull-distant resection cavities (*p* = 0.929).

### 3.3. Volumetric Changes in Resection Cavity, Surrounding Oedema, and FLAIR Hyperintensities

A decrease in resection cavity volume was observed in 57% of patients. Overall, the mean change was −9.1 ± 52.4% (Figure 3A). In 82% of patients, the combined volume of the resection cavity and surrounding oedema decreased, with an overall mean difference of −32.0 ± 29.6% (Figure 3B). The volume of the FLAIR hyperintensities reduced from MRI 1 to MRI 2 in 96% of patients, with an overall mean reduction of −38.0 ± 25.0% (Figure 3C).

### 3.4. Difference in Midline Shift and Subdural Haematoma

The mean postoperative midline shift in MRI 1 was 5.30 ± 4.30 mm. When compared to MRI 2, there was either no change or a decrease, with an overall mean difference of −2.64 ± 2.73 mm (range 0–11 mm). In 68% of patients, the subdural haematoma thickness decreased; in 25% of patients, it increased (7% of patients had no change). Overall, the mean difference in subdural haematoma thickness between MRI 1 and 2 was −1.28 ± 3.74 mm (−10.6 ± 48.7%).

## 4. Discussion

Adjuvant radiotherapy doubles overall survival in GBM patients [3]. Its aim is to improve local tumour control while minimising neurotoxicity. Accurate target volume delineation requires an MRI obtained as close to radiation treatment start as possible (3–5 weeks after surgery), different from the immediate postoperative MRI (24–48 h) [7]. Although recommended and explicitly stated in the majority of study protocols of modern clinical radiotherapy trials in this context, this is not always performed [9,15]. Potential reasons are manifold: limited availability of MRI (especially in patients with pacemakers or similar devices), wanting to prevent treatment delay (e.g., because of planned neurorehabilitation), added costs and resources, problems with insurance, or practical considerations such as patient preference (e.g., claustrophobia). In these cases, radiation treatment planning is based on the target volume as defined by the immediate postoperative MRI, which might impair disease control outcomes. Here, we sought to compare this immediate postoperative MRI with the radiation treatment planning MRI in terms of postoperative neuroanatomical changes, which might influence target volume delineation and subsequent treatment planning, to underscore the value and need of this second MRI

There is no consensus on target volume delineation for high-grade glioma. The most common contouring guidelines recommend starting with the resection cavity and any contrast-enhancing residual tumour, with or without the inclusion of postoperative peritumoral oedema. Then, a generous GTV to CTV margin is added to account for microscopic tumour spread (Table 3). This can result in substantial irradiated volumes, with a subsequent high risk of treatment-related side effects such as neurocognitive impairment or radionecrosis [16,17]. Although multifactorial, total dose, fraction size, and volumetric parameters are thought to be strongly associated with these risks.

Randomised trials are lacking, but there appears to be no difference in relapse rate or pattern of failure depending on the guideline used [19,20]. In order to account for microscopic (low-grade) tumour infiltration, surrounding oedema can be included. This, however, results in significantly higher treatment volumes [19]. Several studies have confirmed that recurrences are mainly at the resection margin (in-field) [21,22,23]. As further dose escalation beyond 60 Gy does not yield improved local control, optimisation of target volume delineation is crucial [24].

In a recent update of the ESTRO-ACROP (now ESTRO-EANO) guideline, a reduced GTV to CTV margin of 15 mm (instead of 20 mm) is recommended, and the inclusion of oedema within the CTV is not advised. T2/FLAIR signal abnormalities, which may represent non-enhancing tumours, should, however, be considered for inclusion within the CTV [9]. Recent evidence even suggests that a further reduction of the GTV to CTV margin to 10 mm is feasible, as the majority of recurrences arise in the resection cavity, and smaller margins lead to a significant reduction of radiation doses to healthy tissues [25]. This should be investigated further in future trials.

Our data suggest short-term changes in postoperative brain anatomy between MRI 1 and 2, with an overall reduction of surrounding oedema and FLAIR hyperintensities. Changes in resection cavity volume showed more variation. Six similar studies investigating MRI-based volumetric changes in the context of target volume delineation and radiation treatment planning for high-grade glioma were identified and are summarised in Table 4. Although MRIs at different points in time are being compared, the overall trend shows marked volumetric changes (mainly decreases) in the resection cavity and subsequent GTV, CTV, and planning target volume (PTV). To the best of our knowledge, the current series is the largest one to date and confirms the findings of these previous reports, implicating that postoperative changes are a dynamic process.

Only a few studies have investigated subsequent changes in relative resection cavity position, which is influenced by both intrinsic (e.g., cavity bleeding) and extrinsic (e.g., the observed difference in midline shift or subdural haematoma up to 11 or 10 mm, respectively) factors. We noted a mean change of 3.04 ± 3.90 mm, with a maximum displacement of 22.1 mm in one patient, whereas the tumour location (lobe) had no impact; our study is the first to identify that there are significantly higher shifts in skull-distant (>25 mm) resection cavity borders, in comparison with skull-near counterparts (≤25 mm). Extensive displacements of the resection cavity could potentially result in a geographic miss if only MRI 1 is used for target volume delineation, as the prescribed dose is not administered to the clinically relevant site (i.e., the resection cavity and surrounding tissue). This phenomenon has also been observed by Manon et al., who compared the immediate postoperative MRI with an MRI during radiation treatment used for boost planning [26]. Here, however, it should be considered that the radiation treatment itself might have influenced this, which could not have been the case in our series (where both MRI 1 and 2 were prior to radiation treatment initiation).

All patients showed a decrease in midline shift between MRI 1 and 2, which is to be expected following surgical resection of an intracranial mass. Of note is the observed residual midline shift in MRI 2 after a median interval of 19.5 days. A further decrease following MRI 2 might thus be possible, which underscores the value of mid-treatment MRI (e.g., for boost planning) [26]. Preoperative midline shift ≥10 mm has been shown to be associated with reduced overall survival after surgery for GBM [27]. The impact of residual postoperative midline shift, however, remains to be elucidated.

**Table 4 cancers-15-02840-t004:** Summary of studies investigating MRI-based volumetric changes in the context of radiation treatment planning for high-grade glioma. MRI = magnetic resonance imaging; WHO = World Health Organisation; RT = radiation therapy; RTOG = Radiation Therapy Oncology Group; n/a = not available; RC = resection cavity; GTV = gross tumour volume; CTV = clinical target volume; PTV = planning target volume; FLAIR = fluid-attenuated inversion recovery.

Author (Year)	*n*	WHO Grade	Timing MRI 1	Timing MRI 2	Contouring Guideline	Outcome
Manon et al. (2004) [26]	15	4 (100%)	<24 h postoperative	<7 days prior to RT boost	RTOG	80% had change in RC and GTV resulting in geographic miss
Shukla et al. (2005) [28]	15	3 (47%) 4 (53%)	1 day before RT	end of RT week 5	RTOG	80% had GTV decrease
Tsien et al. (2005) [29]	21	3 (38%) 4 (62%)	1–2 weeks before RT	RT week 1 and 3	RTOG	89% had GTV change
Champ et al. (2012) [30]	24	3 (33%) 4 (67%)	<48 h postoperative	day of simulation median (range) 17 (7–32) days interval	RTOG	significant changes in GTV (22% volume decrease) and CTV (20% volume decrease)
Yang et al. (2016) [31]	11	2 (36%) 3 (46%) 4 (18%)	before RT	end of RT	RTOG	decrease in RC, GTV, and PTV
Şenkesen et al. (2022) [32]	24	4 (100%)	shortly before RT	shortly before RT boost median (range) 29 (25–38) days interval	RTOG	significant changes in GTV, CTV, and PTV
current series (2023)	28	2 (7%) 4 (93%)	<24–48 h postoperative	before RT median (range) 19.5 (8–50) days interval	n/a	overall decrease in RC, surrounding oedema, and FLAIR

In the current series, we did not assess the dosimetric differences that might have resulted from the observed volumetric changes (i.e., how the target coverage and OAR dose distribution would have been if the radiotherapy was planned with MRI 1 instead of 2). This has, however, been assessed previously. Yang et al. observed better sparing of the OARs if IMRT was replanned with an MRI performed at the end of radiation treatment, mainly due to shrinkage of the resection cavity [31]. Şenkesen et al. came to the same conclusions after MRI-based replanning of the boost volume [32]. The subsequent clinical impact and relevance (e.g., regarding treatment-related side effects, local control, and survival) are, however, still unclear. This should be the subject of future investigations. In this context, careful considerations should be made in order to balance treatment burden and subsequent quality of life impairments versus marginal volumetric, dosimetric, or clinical benefits, given the poor general prognosis of high-grade glioma.

Our trial is not without limitations. The relatively small sample size (although the largest one on this topic to date) warrants caution with a generalisation of the results (especially for subgroup calculations such as the impact of the tumour location). Furthermore, the data presented only applies to patients who were amenable to surgical resection. In those patients receiving only surgical biopsy (e.g., due to patient preference, comorbidity, multifocal disease, high risk of postoperative adverse outcome), it is currently not possible to predict the extent of volumetric changes (fewer haematoma or changes in oedema might be observed, but on the other hand tumour progression before treatment start or even during treatment due to the presence of macroscopic disease cannot be ruled out). This should be the aim of future investigations.

The integration of image-guided radiation therapy (IGRT) into clinical practice might further improve target volume coverage. Marked neuroanatomical changes during radiation treatment (e.g., resorption of subdural haematoma with subsequent regression of midline shift) can be readily visualised with cone beam computed tomography (CBCT), whereas more subtle changes (e.g., in FLAIR hyperintensities) can be assessed using MRI-guided radiotherapy (MRgRT), which might prompt adaptive radiation treatment planning [33,34]. In a prospective comparison, MRgRT resulted in reduced doses of healthy brain tissue [35]. Furthermore, recent advances in diagnostic MRI (e.g., spectroscopy or probabilistic tractography to define the true tumour extent) allow for individualised radiation treatment planning [36,37]. Future trials will establish the exact role and benefit of these features.

Proton therapy might limit the dose to healthy brain structures and is thus of interest as a treatment modality for high-grade glioma, which requires high doses to large areas. Early data suggest similar disease outcomes with better tolerance when compared to conventional photon radiation therapy [38]. Further evidence is, however, needed to confirm these findings and to establish the exact role of proton therapy in this context. Due to the steep dose gradients of proton beams, an accurate delineation of the target volume is even more important in proton therapy, as even minimal shifts can greatly increase the risk of a geographic miss.

## 5. Conclusions

GBM holds a poor prognosis, meaning that every treatment step should be optimised to maximise survival chances. Our data support the use of an MRI to aid in target volume delineation performed as close to radiation treatment start as possible, as there are marked differences with the immediate postoperative MRI. Planning adapted to the actual anatomy at the time of radiation improves target coverage, which limits the risk of geographic miss and thus optimises outcome. Furthermore, reduced doses to healthy tissues are expected, which minimises treatment-related toxicity in patients undergoing adjuvant radiation for high-grade glioma.

## Figures and Tables

**Figure 1 cancers-15-02840-f001:**
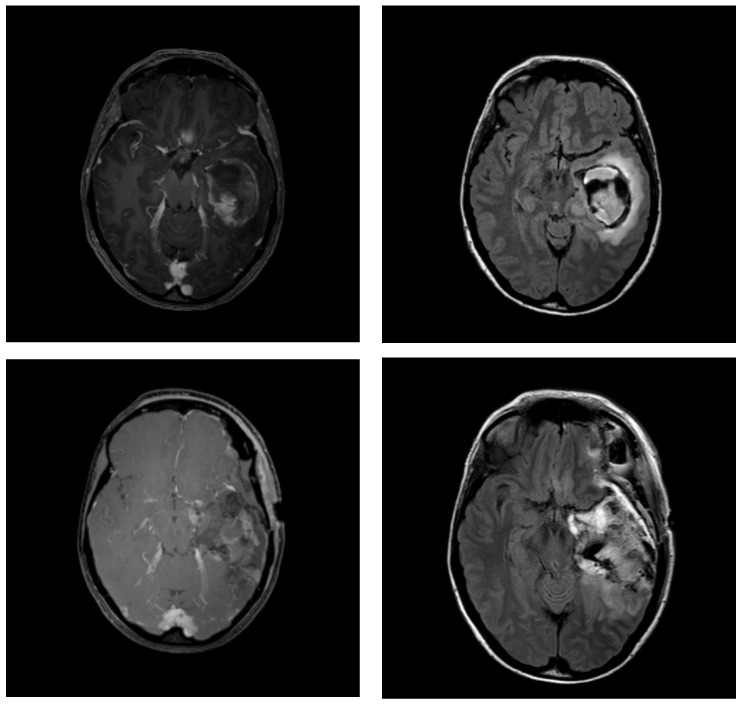
Example MRI images of a patient with temporal glioblastoma: (**top**) 2 days before surgery, (**middle**) 2 days after surgery, and (**bottom**) 17 days after surgery (for radiation treatment planning). T1-weighted contrast-enhanced (Gadolinium) images on the left and T2-weighted FLAIR images on the right. MRI = magnetic resonance imaging; FLAIR = fluid-attenuated inversion recovery.

**Figure 2 cancers-15-02840-f002:**
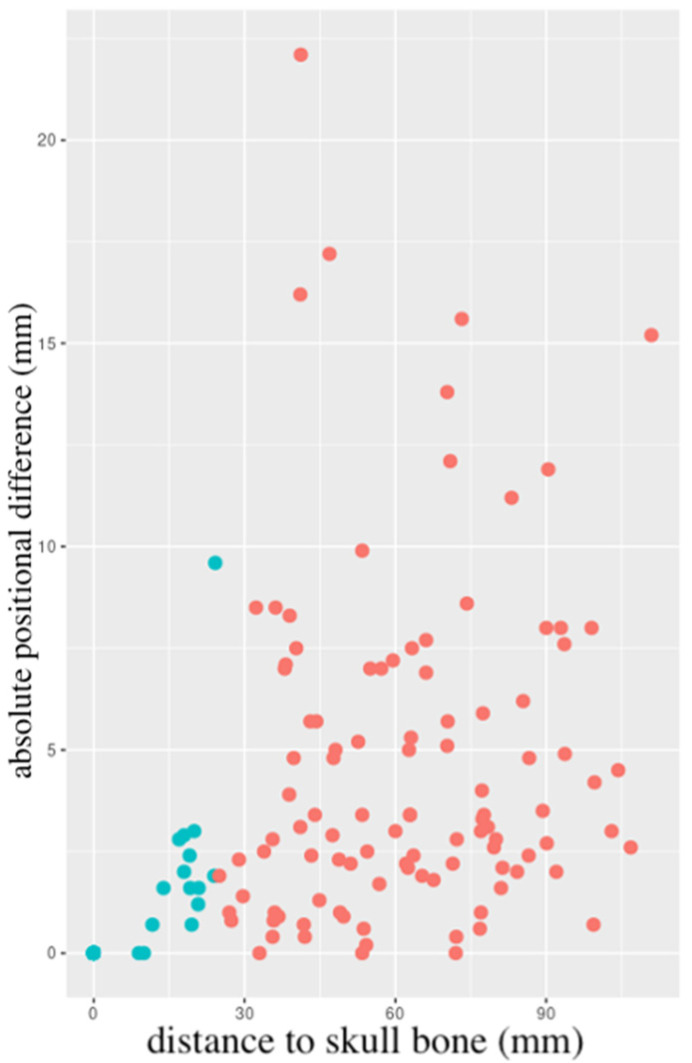
Scatter plot of the individual relative positional changes between MRI 1 and 2, depending on the initial (MRI 1) distance to the skull bone. Skull-near (≤25 mm) resection cavity borders in blue, skull-distant (>25 mm) in red. MRI = magnetic resonance imaging.

**Figure 3 cancers-15-02840-f003:**
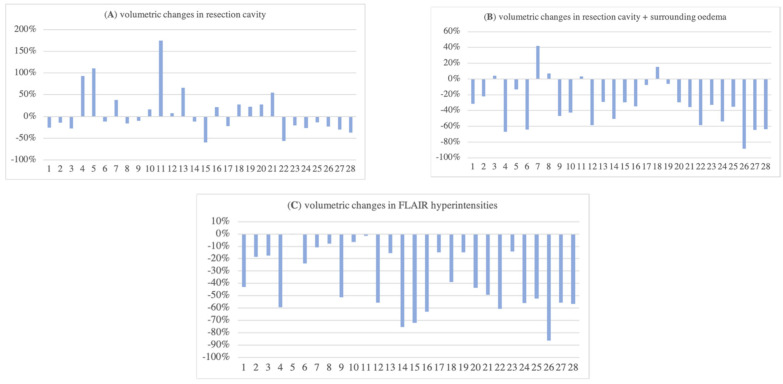
Relative volumetric changes in resection cavity without (**A**) or with (**B**) surrounding oedema as well as FLAIR hyperintensities (**C**) between MRI 1 and 2. MRI = magnetic resonance imaging; FLAIR = fluid-attenuated inversion recovery.

**Table 1 cancers-15-02840-t001:** General patient characteristics (*n* = 28). MRI = magnetic resonance imaging.

	*n* (%)
median age (range) in years	61 (26–78)
sex	
male	18 (64)
female	10 (36)
diagnosis	
glioblastoma	26 (93)
oligodendroglioma	2 (7)
tumour location	
frontal lobe	13 (46)
parietal lobe	7 (25)
temporal lobe	7 (25)
occipital lobe	1 (4)
interval MRI 1–2 (range) in days	19.5 (8–50)

**Table 2 cancers-15-02840-t002:** Mean absolute difference in resection cavity position relative to the skull bone between MRI 1 and 2. The difference between skull-near and skull-distant resection cavity borders was statistically significant (*p* < 0.001), even after accounting for the interval between MRI 1 and 2. MRI = magnetic resonance imaging; SD = standard deviation.

	Total	Skull-Near (≤25 mm)	Skull-Distant (>25 mm)
number of measurements (*n*)	168	64	104
mean difference ± SD (mm)	3.04 ± 3.90	0.50 ± 1.40	4.61 ± 4.12
minimum (mm)	0	0	0
maximum (mm)	22.1	9.6	22.1

**Table 3 cancers-15-02840-t003:** Common contouring guidelines for high-grade glioma. GTV = gross tumour volume; CTV = clinical target volume; PTV = planning target volume; ESTRO-ACROP = European Society for Radiotherapy and Oncology—Advisory Committee for Radiation Oncology Practice; RTOG = Radiation Therapy Oncology Group.

Reference	Dose	GTV	CTV	PTV
ESTRO-ACROP [8]	60 Gy	T1 + cavity	+2 cm	+3–5 mm
RTOG [18]	46 Gy 16 Gy	GTV_1_ = T1 + cavity + T2 GTV_2_ = T1 + cavity	+2 cm	+3–5 mm

## Data Availability

All data are available in Appendix A.

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
