# Peer review of "Impact of Postoperative Changes in Brain Anatomy on Target Volume Delineation for High-Grade Glioma"

_cancers, 2023, doi:10.3390/cancers15102840_

Round 1

Reviewer 1 Report

Dejonckheere et al., discuss the importance of utilizing MRI scans within close proximity to radiation start. This is well-known by most CNS-focused radiation oncologists.

1.     The authors should elaborate on barriers and practical limitations to obtaining another MRI for radiation therapy. As provided in table 4, this has been evaluated by others. Additionally, the need to obtain a new MRI following time from resection is stipulated in recent/accruing NRG trials, which was not discussed.

2.     Briefly discussed in the introduction are barriers to not obtaining a new MRI prior to radiotherapy e.g.: insurance, need to start adjuvant treatment quickly due to rehabilitation stays, difficulty obtaining a new MRI due to availability. This should be elaborated on in the discussion as it is a key tenant of balancing practicality versus obtaining optimal planning scans

3.     The group hypothesizes “fewer volumetric changes” in glioblastomas that were only biopsied: please provide data either in support or against this statement.  Given the rapid growth of these tumors, and duration of radiotherapy, it is quite possible that there is progression between biopsy and chemoradiation and/or during treatment.

4.     The authors note “no consensus on target volume delineation for high-grade glioma” and cite references 8,14-16. At least one reference should be updated: PMID: 37059335. Additionally, many experts (as discussed in this reference, as well as others: PMID: 36529439) have decreased CTV expansions to 1-1.5 cm. Additionally, despite knowledge that FLAIR regions contain glioma cells, some European centers do not regularly include FLAIR in their treatment areas as the majority of recurrences are in the resection bed. Please include discussion on these points as decreased volumes also decrease toxicity.

5.     The aim of this manuscript is to support the role of an MRI to aid in target volume delineation as close to radiation treatment start as possible. Literature already exists to support this. As the main argument for this is to reduce toxicity, the authors should provide dosimetric comparisons utilizing MRI 1 and MRI 2. The authors note they did not perform these comparisons, however this addition will greatly strengthen the paper and add a degree of novelty.

6.     The authors should expand on whether literature demonstrates using older MRIs affects local control or survival. This is another consideration that can greatly strengthen their paper. Conversely, in the context of the overall survival of glioblastoma patients, treatment package time may be more important than newer imaging for a dosimetric and/or acute toxicity benefit

7.     The impact of resection cavity changes in the context of proton therapy should be discussed

Author Response

Dejonckheere et al., discuss the importance of utilizing MRI scans within close proximity to radiation start. This is well-known by most CNS-focused radiation oncologists.

The authors would like to thank the reviewer for carefully reading through the manuscript and providing useful feedback and recommendations. Our responses are stated below, in a point-by-point fashion.

  1. The authors should elaborate on barriers and practical limitations to obtaining another MRI for radiation therapy. As provided in table 4, this has been evaluated by others. Additionally, the need to obtain a new MRI following time from resection is stipulated in recent/accruing NRG trials, which was not discussed.
  2. Briefly discussed in the introduction are barriers to not obtaining a new MRI prior to radiotherapy e.g.: insurance, need to start adjuvant treatment quickly due to rehabilitation stays, difficulty obtaining a new MRI due to availability. This should be elaborated on in the discussion as it is a key tenant of balancing practicality versus obtaining optimal planning scans.

Thank you for these suggestion. These barriers to performing a second MRI are indeed very relevant to the manuscript. We thus expanded this part to elaborate more on the reasons behind it (1st paragraph of the discussion; lines 181-195). The recent ESTRO-EANO guideline update was also added as a reference (line 186).

  1. The group hypothesizes “fewer volumetric changes” in glioblastomas that were only biopsied: please provide data either in support or against this statement. Given the rapid growth of these tumors, and duration of radiotherapy, it is quite possible that there is progression between biopsy and chemoradiation and/or during treatment.

Thank you for this comment. We do not have proof to back this claim, and the point of progression during treament as raised by the reviewer is indeed plausible. The statement was therefore altered, to reflect a more balanced conclusion in this unexplored setting of biopsied patients and a future perspective was added (lines 278-282).

  1. The authors note “no consensus on target volume delineation for high-grade glioma” and cite references 8,14-16. At least one reference should be updated: PMID: 37059335. Additionally, many experts (as discussed in this reference, as well as others: PMID: 36529439) have decreased CTV expansions to 1-1.5 cm. Additionally, despite knowledge that FLAIR regions contain glioma cells, some European centers do not regularly include FLAIR in their treatment areas as the majority of recurrences are in the resection bed. Please include discussion on these points as decreased volumes also decrease toxicity.

We thank the reviewer for suggesting this very recent and relevant reference of Niyazi et al., which we added to the discussion (lines 215-218). We chose, however, not to omit the ESTRO-ACROP guideline, as is was used for target volume delineation in this trial and, furthermore, we reason that radiation oncologist will currently still be most familiar with the “older” guideline. The differences between both guidelines are, however, highlighted.

The excellent work by Minniti and colleagues is also highly relevant to our manuscript and was thus added and discussed (lines 219-222).

  1. The aim of this manuscript is to support the role of an MRI to aid in target volume delineation as close to radiation treatment start as possible. Literature already exists to support this. As the main argument for this is to reduce toxicity, the authors should provide dosimetric comparisons utilizing MRI 1 and MRI 2. The authors note they did not perform these comparisons, however this addition will greatly strengthen the paper and add a degree of novelty.

Thank you for this comment. We agree that a dosimetric comparison would be of great added value to this area of research, as only few trials have investigated this. In the current manuscript, we chose to focus on MRI-based volumetric changes in the context of radiation treatment planning as opposed to dosimetric differences. The two trials that have investigated this in the past are referenced (Yang et al. and Åženkesen et al.). The clinical relevance of these changes has never been reported before. A sentence was added to this paragraph to emphasise the need for future investigations on this topic (line 269).

  1. The authors should expand on whether literature demonstrates using older MRIs affects local control or survival. This is another consideration that can greatly strengthen their paper. Conversely, in the context of the overall survival of glioblastoma patients, treatment package time may be more important than newer imaging for a dosimetric and/or acute toxicity benefit.

Thank you for this comment. As discussed under point 5, the clinical relevance of these volumetric/dosimetric changes has never been investigated before (which was added as a future perspective; line 269). The second point raised, is of course very relevant to this discussion, and was added to this paragraph (lines 268-272).

  1. The impact of resection cavity changes in the context of proton therapy should be discussed.

As suggested by the reviewer, the impact of resection cavity changes is even more important in the context of proton therapy. A paragraph was added to the discussion, to highlight this topic (lines 294-301).

Reviewer 2 Report

The manuscript titled "Impact of postoperative changes in brain anatomy on target volume delineation for high-grade glioma" provides important insights into the need for an MRI in radiation treatment planning for high-grade glioma. The authors compared the structures required for target volume delineation between immediate postoperative MRI and radiation treatment planning MRI in 28 patients. They found that there were marked short-term volumetric changes in the resection cavity, surrounding oedema, and FLAIR hyperintensity volumes, and a mean change in resection cavity position. These findings support the role of MRI in aiding target volume delineation as close to radiation treatment start as possible.

Overall, the manuscript is well-written and the methodology is sound. The study design is appropriate and the sample size is adequate for the purpose of the study. The results are presented clearly and the implications for clinical practice are well-discussed. However, the authors should provide more information on the limitations of the study, particularly in terms of generalizability and the potential impact of other factors such as surgery type or adjuvant therapies.

Some suggested points before final approval of the manuscript are:

1.      The introduction could be revised to provide more context on the importance of malignant glioma and the significance of adjuvant radiotherapy for the management of the disease.

2.      The discussion section could be expanded to include a more detailed analysis of the potential reasons why immediate postoperative MRI is used for radiation treatment planning, despite the benefits of obtaining an MRI closer to the start of radiation treatment.

3.      The section on target volume delineation could be revised to provide more information on the risks of treatment-related side effects associated with high irradiated volumes.

4.      The section on common contouring guidelines for high-grade glioma could be revised to provide more information on the differences between the ESTRO-ACROP and RTOG guidelines, and to explain the rationale for including the surrounding edema in the target volume.

5.      The authors can discuss the current state of research on novel diagnostic and therapeutic approaches in glioma.  (eg. Role of exosomes in malignant glioma: MicroRNAs and proteins in pathogenesis and diagnosis, Neurosphere and adherent culture conditions are equivalent for malignant glioma stem cell lines)

6.      The section on volumetric changes in postoperative brain anatomy could be revised to provide more information on the implications of these changes for target volume delineation and radiation treatment planning.

7.      The section on subsequent changes in relative resection cavity position could be revised to provide more information on the potential impact of these changes on treatment outcomes, and to explain why extensive displacements of the resection cavity could potentially result in geographic miss if only MRI 1 is used for target volume delineation.

Author Response

The manuscript titled "Impact of postoperative changes in brain anatomy on target volume delineation for high-grade glioma" provides important insights into the need for an MRI in radiation treatment planning for high-grade glioma. The authors compared the structures required for target volume delineation between immediate postoperative MRI and radiation treatment planning MRI in 28 patients. They found that there were marked short-term volumetric changes in the resection cavity, surrounding oedema, and FLAIR hyperintensity volumes, and a mean change in resection cavity position. These findings support the role of MRI in aiding target volume delineation as close to radiation treatment start as possible.

Overall, the manuscript is well-written and the methodology is sound. The study design is appropriate and the sample size is adequate for the purpose of the study. The results are presented clearly and the implications for clinical practice are well-discussed. However, the authors should provide more information on the limitations of the study, particularly in terms of generalizability and the potential impact of other factors such as surgery type or adjuvant therapies.

Some suggested points before final approval of the manuscript are:

The authors would like to thank the reviewer for carefully reading through the manuscript and providing useful feedback and recommendations. We aimed to elaborate on limitations and future perspectives. Our responses are stated below, in a point-by-point fashion.

  1. The introduction could be revised to provide more context on the importance of malignant glioma and the significance of adjuvant radiotherapy for the management of the disease.

The importance of malignant glioma was highlighted by adding the global prevalence (line 45). The significance of adjuvant radiotherapy in its management was also added by emphasising the impact on overall survival (line 47).

  1. The discussion section could be expanded to include a more detailed analysis of the potential reasons why immediate postoperative MRI is used for radiation treatment planning, despite the benefits of obtaining an MRI closer to the start of radiation treatment.

Thank you for this suggestion. These barriers to performing a second MRI are indeed very relevant to the manuscript. We thus expanded this part to elaborate more on the reasons behind it (1st paragraph of the discussion; lines 181-195).

  1. The section on target volume delineation could be revised to provide more information on the risks of treatment-related side effects associated with high irradiated volumes.

Thank you for this excellent recommendation. This paragraph was expanded to include other known risk factors for such treatment-related side effects (lines 202-203).

  1. The section on common contouring guidelines for high-grade glioma could be revised to provide more information on the differences between the ESTRO-ACROP and RTOG guidelines, and to explain the rationale for including the surrounding edema in the target volume.

Thank you for this recommendation. The differences between the two guidelines were highlighted (lines 196-199), as was the rationale for including peritumoural oedema in the RTOG guideline (lines 209-211).

  1. The authors can discuss the current state of research on novel diagnostic and therapeutic approaches in glioma. (eg. Role of exosomes in malignant glioma: MicroRNAs and proteins in pathogenesis and diagnosis, Neurosphere and adherent culture conditions are equivalent for malignant glioma stem cell lines).

Thank you for this comment. We added a discussion of emerging evidence on further reduction of the GTV to CTV margin (lines 219-222), as well as a paragraph on the role and limitations of proton therapy in this context (lines 294-301). The role of exosomes and associated topics are beyond the scope of the current manuscript, which focusses mainly on imaging for radiation treatment planning.

  1. The section on volumetric changes in postoperative brain anatomy could be revised to provide more information on the implications of these changes for target volume delineation and radiation treatment planning.
  2. The section on subsequent changes in relative resection cavity position could be revised to provide more information on the potential impact of these changes on treatment outcomes, and to explain why extensive displacements of the resection cavity could potentially result in geographic miss if only MRI 1 is used for target volume delineation.

Thank you for these suggestions. The two trials that have investigated dosimetric changes in the past are referenced (Yang et al. and Åženkesen et al.). The clinical relevance of these changes has never been reported before. This paragraph was expanded to emphasise the need for future investigations on this topic (lines 268-272) and the concept of geographic miss was elaborated (lines 241-242).

Round 2

Reviewer 1 Report

The timely reply and explanations are appreciated